# Controllable multiple-step configuration transformations in a thermal/photoinduced reaction

Meng-Fan Wang [1], Yan Mi [2], Fei-Long Hu [2✉], Hajime Hirao [3✉], Zheng Niu [1✉], Pierre Braunstein[4] & Jian-Ping Lang [1✉]

Solid-state photochemical reactions of olefinic compounds have been demonstrated to represent powerful access to organic cyclic molecules with specific configurations. However, the precise control of the stereochemistry in these reactions remains challenging owing to complex and fleeting configuration transformations. Herein, we report a unique approach to control the regiospecific configurations of C $=$ C groups and the intermediates by varying temperatures in multiple-step thermal/photoinduced reactions, thus successfully realizing reversible ring closing/opening changes using a single-crystal coordination polymer platform. All stereochemical transitions are observed by in situ single-crystal X-ray diffraction, powder X-ray diffraction and infrared spectroscopy. Density functional theory calculations allow us to rationalize the mechanism of the synergistic thermal/photoinduced transformations. This approach can be generalized to the analysis of the possible configuration transformations of functional groups and intermediates and unravel the detailed mechanism for any inorganic, organic and macromolecular reactions susceptible to incorporation into single-crystal coordination polymer platforms.

[1] College of Chemistry, Chemical Engineering and Materials Science, Soochow University, Suzhou, Jiangsu, People's Republic of China. [2] Guangxi Key Laboratory of Chemistry and Engineering of Forest Products, Guangxi University for Nationalities, Nanning, Guangxi, People's Republic of China. [3] School of Life and Health Sciences, The Chinese University of Hong Kong, Shenzhen, Longgang Dist., Shenzhen, Guangdong, People's Republic of China. [4] Université de Strasbourg - CNRS, Institut de Chimie (UMR 7177 CNRS), Strasbourg, France. ✉email: hflphd@163.com;hirao@cuhk.edu.cn;zhengniu@suda.edu.cn; jplang@suda.edu.cn

Stereochemical control in the construction of specific organic compounds is a persistent challenge in contemporary synthetic chemistry[1,2]. Over the past decades, [2 + 2] photocycloaddition reactions have been widely applied to the synthesis of specific cyclobutanes because they represent high yielding single transformations. However, cyclobutanes have four stereocenters, and their control suffers from relatively poor regio- and stereoselectivity[3–9]. Various supramolecular templates[10–12], host-guest assemblies[13,14], quantum dots[15], chiral molecular catalysts[16–18], and Lewis acid cocatalysts[19,20] have been developed to control the enantioselectivity and diastereoselectivity of [2 + 2] photocycloaddition transformations. Although it is possible to capture cyclic intermediates at low temperature[21], the control of their stereochemistry during the reaction has not yet been documented. This would be highly desirable but also very challenging.

Benefiting from their diversity, designability and crystallinity, coordination polymers (CPs)[22–26] have been used as platforms to realize directed photocycloaddition reactions. While strategies have been applied in photocatalytic reactions to construct stereoselectively monocyclobutanes[27–31], the stereochemical control of multicyclic compounds obtained by photocycloaddition of dienes and polyenes remains challenging[32]. When targeting multicyclic molecules by solid-state photocycloaddition reactions, a precise setting of the configuration of the olefinic reactants is key to controlling the product stereochemistry[33–37]. The prearrangement of the C = C groups could be influenced by multiple factors, such as the presence of specific metal ions, auxiliary ligands, solvents, or substituents of olefinic reactants[38–40]. Only a few examples have reported the temperature as a tool to adjust the mutual arrangement of the C = C groups[41], yielding different cyclic products in the photocycloaddition reaction[35]. We suggest that if a solid-state photocycloaddition reaction could proceed at different temperatures, various arrangements of the olefinic reactants could result and lead to intermediates with different configurations. This could facilitate our understanding of the complex stereochemical features occurring during photocycloaddition.

Single-crystal CPs appear to be potentially powerful platforms for tracking the stereo-chemical structure of reaction intermediates[42]. Many structural characterization techniques, especially single-crystal X-ray diffraction (SCXRD)[43,44], are available but in most cases, single crystals of CPs do not withstand the internal stress resulting from external stimuli and lose the single-crystal character required for SCXRD investigation[45–47]. Exploring the stereochemical structures of intermediates involved in a multiple-step thermal/photoinduced reaction using a single-crystal CP platform appears highly attractive but challenging.

In this work, using a single-crystal CP platform, we describe for the first time controllable configuration transformations in the course of a multiple thermal/photoinduced reaction. The diolefin ligand 4,4′-(5-fluoro-1,3-phenylene)bis-(ethene-2,1-diyl)) dipyridine (F-1,3-bpeb) is used to synthesize CP [Cd$_2$(F-1,3-bpeb)$_2$(3,5-DBB)$_4$] (CP1, 3,5-HDBB = 3,5-dibromobenzoic acid). Two diene isomers, 4,4′-(3,4-bis(3-fluoro-5-(2-(pyridin-4-yl)vinyl)phenyl)cyclobutane-1,2-diyl)dipyridine (1 and 1′), and two isomeric metacyclophanes, syn-3,4,12,13-tetrakis(4-pyridyl)-8,17-bisfluoro-1,2,9,10-diethano[2.2]metacyclophane (2α and 2β), are obtained after photoirradiation and/or thermal treatment of the F-1,3-bpeb pairs in single-crystal CP1 (Supplementary Fig. 1). Interestingly, the original configuration of the monomer in the initial CP1 is altered at different temperatures during the thermal/photoinduced reaction process. As shown in Fig. 1, the synthesis of CP1-2α from CP1 requires a change in the configuration of three C = C double bonds of two opposite ligands in CP1, which implies going through three or more steps and isomeric intermediates. The good crystallinity of CP1 allows us to

set different temperature domains to trace the changes in the configuration of the isomers in the synergistic thermal/photoinduced reactions using SCXRD and other techniques. Although the crystal used has cracks, its morphology and transparent appearance are retained in the multiple-step transformations. By investigating the stepwise changes and collecting the snapshots in a series of thermal/photoinduced reactions, the configurations of C = C groups and the reversible ring-closing/opening changes in a dicyclobutane system are revealed, which allow the use of temperature to control the whole stereochemical process. The proposed mechanism is further supported by density functional theory (DFT) calculations.

## Results

**Synthesis and characterization of CP1.** The Cd(II)-based coordination polymer [Cd$_2$(F-1,3-bpeb)$_2$(3,5-DBB)$_4$] (CP1) was isolated from solvothermal reactions of 3CdSO$_4$·8H$_2$O with F-1,3-bpeb and 3,5-HDBB. SCXRD analysis revealed that CP1 contains dinuclear Cd units linked by pairs of F-1,3-bpeb to give a one-dimensional (1D) zigzag chain structure. As shown in Fig. 2, the configurations of the C = C groups in two opposite F-1,3-bpeb molecules are different, resulting in one parallel and one crisscross arrangement of C = C bond pairs in the chain. The distance between the two parallel C = C groups is 3.82 Å, which, according to Schmidt's rule[48], should allow the photocycloaddition reaction to occur. The separation between the crossed C = C pairs in CP1 is 3.692 Å, but a photocycloaddition reaction requires that one of the C = C groups rotates to the parallel position (Fig. 2 and Supplementary Fig. 2).

**[2 + 2] photocycloaddition reactions of CP1.** The arrangement of the C = C bonds in CP1 should yield a monocyclobutane product upon UV light irradiation. However, dicyclobutane was obtained (λ = 365 nm) after 1 h at 25 °C. SCXRD analysis revealed that the basic 1D zigzag chain structure in the crystal is almost the same as that of CP1, and is composed of dinuclear Cd units with an exo,endo-dicyclobutane (2β), giving [Cd$_4$(2β)$_2$(3,5-DBB)$_8$] (CP1-2β) (Fig. 2). The NMR and mass spectra of the isolated ligand from decomposed crystals reveal full conversion of the monomer F-1,3-bpeb to 2β (Supplementary Figs. 9, 10, 20). An X-ray analysis confirmed the stereostructure of 2β after recrystallization (Supplementary Fig. 3). Formation of the latter suggests that one of the C = C groups in the crisscross pair in CP1 undergoes a pedal motion to give parallel C = C bond pairs under UV irradiation, thus allowing the formation of CP1-2β. To investigate the influence of temperature on the rotation of the C = C groups of CP1, differential scanning calorimetry (DSC) was performed and showed no significant peaks until 260 °C, which indicated that the rotation of C = C groups in CP1 cannot be captured in the absence of irradiation (Supplementary Fig. 24a). These results suggest that the synergistic effect of temperature and UV radiation is likely to be the key to the configuration transformation of the C = C groups.

**Controllable multiple-step configuration transformations.** The established rotation of the crisscross arranged olefin pair under UV irradiation triggered our interest in the possible isolation of different cyclic isomers with specific configurations by adjusting the reaction temperature. At 208 °C, a new CP [Cd$_2$(2α)(3,5-DBB)$_4$] (CP1-2α) was obtained after 3 h of irradiation under 365 nm UV light (Fig. 2 and Supplementary Fig. 13). The structure of CP1-2α, determined by SCXRD, contains a 1D zigzag chain constructed of 2α and dinuclear Cd units (Fig. 2). In contrast to 2β, 2α includes an exo,exo-dicyclobutane rather than an exo,endo-dicyclobutane as in 2β, as confirmed by SCXRD

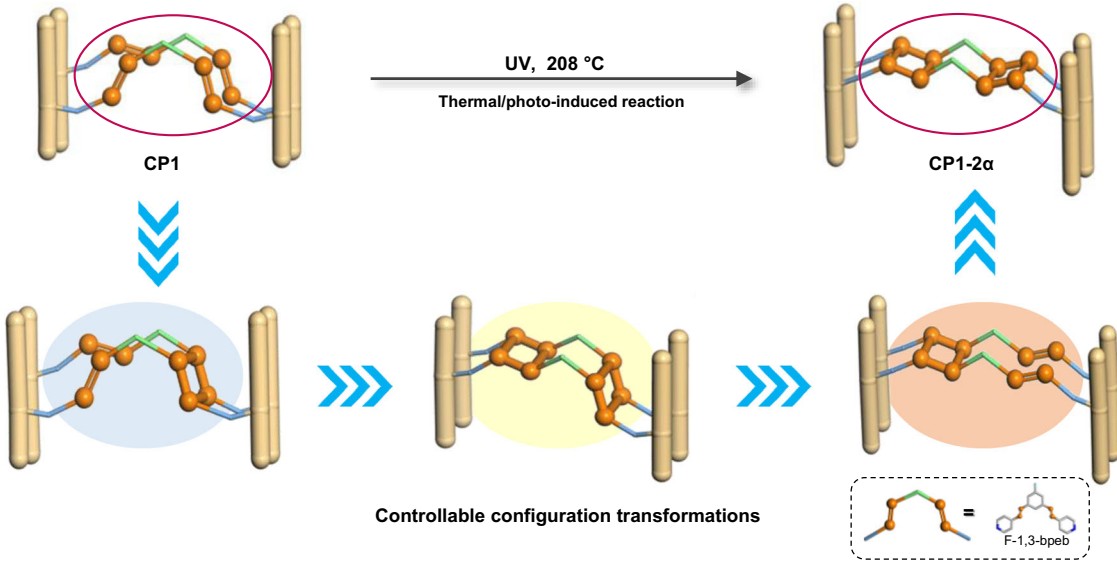

**Fig. 1 Schematic illustration.** The controllable multiple-step configuration transformations of **CP1** to **CP1**-**2α** by adjusting the reaction temperatures, which require going through four steps and three isomeric intermediates. Color codes: N, blue; F, sky blue; O, red; C, gray. Hydrogen atoms are omitted for clarity. All the configurations and transformations of C=C groups associated with the above reactions are highlighted in orange.

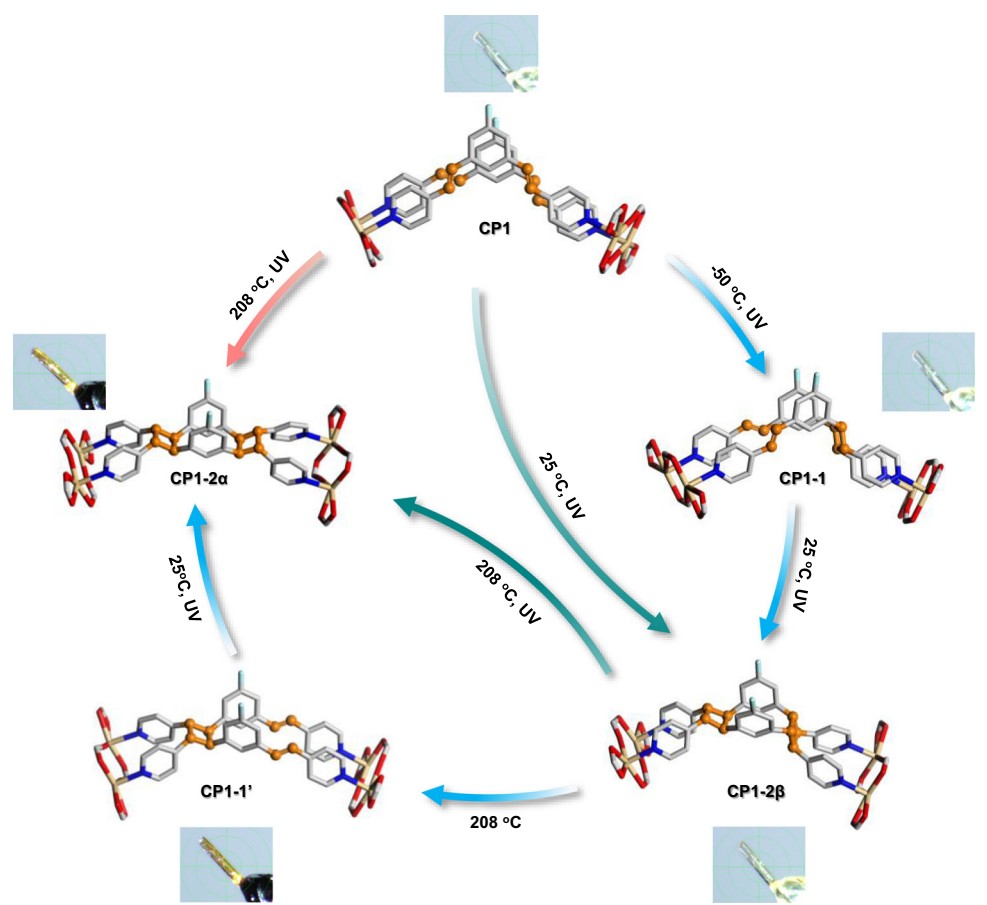

**Fig. 2 Synthesis routes and structures of the multiple-step thermal/photo-induced transformations of CP1 to CP1-2α.** The configurations of **CP1**-**1**, **CP1**-**2β**, **CP1**-**1′**, and **CP1**-**2α** illustrate the monomer transformations for each step. Inset: crystal photographs of **CP1** in the multiple-step thermal/photo-induced transformations. Color codes: Cd, pale yellow; N, blue; F, sky blue; O, red; C, gray. Hydrogen atoms are omitted for clarity. All the configurations and transformations of C=C groups associated with the above reactions are highlighted in orange.

analysis of recrystallized **2α** (Supplementary Fig. 4) and its NMR and MS spectra (Supplementary Figs. 14, 19). Going from **CP1** to **CP1-2α** requires a complex rotation of the C $=$ C groups where the C $=$ C groups with endo-configuration from two opposite ligands in **CP1** end-up in an exo-configuration (Fig. 2). The whole process involves multiple rotation steps and we aimed to use SCXRD to capture snapshots of the intermediates and unravel their molecular structures and the details of the associated configuration changes.

Irradiation of **CP1** was carried out under UV light at −50 °C to investigate the intermediates on the way to **CP1-2β**. The $^1$H and $^{19}$F NMR data indicated the formation of a pure monocyclobutane (Supplementary Fig. 7). The SCXRD analysis revealed that [Cd$_2$(**1**)(3,5-DBB)$_4$] (**CP1-1**) forms a 1D zigzag chain with **1** and dinuclear Cd units (Fig. 2). As expected, only the C $=$ C groups arranged in a parallel manner allowed dimerization while those arranged in a crisscross manner remained intact, since the molecular rotation under UV irradiation was blocked at low temperature.

When the temperature was increased from −50 °C to 25 °C, one of the C $=$ C groups in **CP1-1** rotated to become parallel to another, allowing the formation of **CP1-2β** under UV irradiation. Again, SCXRD indicated that the crisscross C $=$ C pair remained intact and that only the parallel C $=$ C groups became photocoupled. **CP1-1** can be viewed as an intermediate during the formation of **CP1-2β** and this transformation establishes the synergistic effect of UV irradiation and temperature (Fig. 2). A white powder of **1** was isolated by extraction from **CP1-1**, as confirmed by $^1$H and $^{19}$F NMR (Supplementary Fig. 8).

The different structures of **CP1-2β** and **CP1-2α** suggest that breaking of the cyclobutane ring, configuration change of C $=$ C groups and reformation of a new cyclobutane ring occurred. Since a temperature increase can restore the C $=$ C pairs by breaking cyclobutane rings, as inferred from measurements (Supplementary Fig. 24), the transformation of **CP1-2β** to **CP1-2α** was investigated at 208 °C by in situ SCXRD. Heating **CP1-2β** for 4 h at 208 °C afforded pure [Cd$_2$(**1'**)(3,5-DBB)$_4$] (**CP1-1'**). The in-situ SCXRD results showed that one endo-configuration of two cyclobutane rings in **2β** was opened to generate **CP1-1'** with parallel C $=$ C pairs while the other remained in exo-configuration, which was confirmed by $^1$H and $^{19}$F NMR (Fig. 2 and Supplementary Figs. 11 and 12). The configuration of the newly formed C $=$ C pairs is exo, while that of the original cyclobutane is endo. These results indicate that the exo-configuration of the C $=$ C groups is more stable than the endo-configuration at high temperature (Supplementary Fig. 24).

Exposure of **CP1-1'** to UV light at 25 °C for 1.5 h led to the generation of cyclobutane and complete transformation to **CP1-2α**, as confirmed by SCXRD analysis and NMR spectra (Fig. 2). We can thus conclude that the formation of **CP1-2α** goes through the intermediacy of **CP1-1'**.

Remarkably, $^{19}$F NMR of the ligand indicated that **CP1-2β** is the main product when **CP1** was irradiated under UV light at 208 °C for 5 min, and it gradually converts to **CP1-2α** with time (Supplementary Fig. 15). These data suggest that **CP1-2α** is a thermodynamic product while **CP1-2β** is the kinetic product that transforms to **CP1-2α** under UV light irradiation at higher temperatures.

The occurrence of photocycloaddition and cyclo-cleavage reactions in the transformation from **CP1** to **CP1-2α** was monitored by in-situ power X-ray diffraction (PXRD) and in-situ IR analyses. As shown in Fig. 3a, sharp pristine diffraction peaks gradually appeared at $2\theta = 10.7°$, 16.3°, and 22.7°, which are consistent with the patterns of **CP1-1**. Then, the sharp pristine diffraction peaks of **CP1-1** at $2\theta = 11.3°$ diminished and

almost completely disappeared during UV irradiation for 40 min, while new peaks gradually appeared at $2\theta = 10.5°$, 12.8°, 16.7°, and 24.7°, which are consistent with the patterns of **CP1-2β** (Fig. 3b and Supplementary Fig. 22b). Heating the crystals of **CP1-2β** to 208 °C led to a decrease in the sharp pristine diffraction peaks of **CP1-2β** at $2\theta = 5.3°$, 6.6°, 10.6°, 16.4°, and 16.7°, which almost completely disappeared after heating for 4 h, while new peaks gradually appeared at $2\theta = 5.8°$, 6.8°, 11.7°, and 17.6° corresponding to **CP1-1'** (Fig. 3c and Supplementary Fig. 22c). Since no bond rotation occurred during the transformation of **CP1-1'** to **CP1-2α**, no significant change was observed by in-situ PXRD (Supplementary Fig. 22d).

Similar to the in-situ PXRD, the FT-IR study also revealed the occurrence of photocycloaddition and its reverse in the transformation of **CP1** to **CP1-2α**. As shown in Fig. 3d, FT-IR spectra clearly showed the appearance of new cyclobutane rings and the depletion of the vinylene links. Upon exposure of **CP1** to UV light, the intensity of the C-H bending vibration of the olefin groups at approximately 951 cm$^{-1}$ rapidly decreased, while the C–H stretching of cyclobutane rings at 2927 cm$^{-1}$ appeared. Meanwhile, the C $=$ C stretching vibration at 1609 cm$^{-1}$ generally decreased under UV light and increased under thermal conditions, thereby illustrating the transformation of **CP1** to **CP1-2α** (Fig. 3d and Supplementary Fig. 25).

**Theoretical calculations**. To better understand the detailed mechanism of the transformation process from **CP1** to **CP1-2α**, theoretical calculations were performed (for details, see the SI). Our DFT model is based on the single crystal X-ray structure of **CP1**, while the Cd and O atoms are held fixed during the geometry optimization calculations (Supplementary Fig. 26). As shown in Fig. 4a, the barrier for the thermal C–C bond formation of **Int1**(S0) via **TS1**(S0) having an open-shell character is 52.9 kcal/mol, which is too high. Therefore, **Int1**(S0) will first undergo photoexcitation, and the C–C bond formation may occur on the T$_1$ energy surface, to form **Int3** (**CP1-1**). Given that $^1$**Int3** is thermodynamically favored over $^1$**Int4**, a likely pathway involves photoexcitation of $^1$**Int3**, and the rotation and C–C bond formation events then occur in the triplet state, before forming $^1$**Int6** (**CP1-2β**) (Fig. 4b).

The thermal cleavage of two C–C bonds of the cyclobutane of **CP1-2β** is a difficult process, involving a high energy barrier of 36.7 kcal/mol (Fig. 4c). However, this barrier may be overcome at 208 °C to form $^1$**Int8** and then $^1$**Int10** (**CP1-1'**). Although $^1$**Int10** is thermodynamically less favorable than $^1$**Int8**, its formation will be facilitated at high temperatures. Although this thermal C–C bond cleavage occurs in the singlet spin state, the transition state ($^1$**TS4**) has an open-shell character, causing a homolytic cleavage of the C–C bonds. If this C–C bond cleavage event occurs in the triplet state, the process would involve spin inversions. Furthermore, $^3$**TS4** is less stable than $^1$**TS4**. These factors make it unlikely that the triplet pathway is used under thermal conditions. In the final step, $^1$**Int10** (**CP1-1'**) undergoes photoexcitation to form $^3$**Int10**, and the first and second C–C bond formation events occur in the triplet and singlet states, respectively, thus generating the final product $^1$**Int12** (**CP1-2α**).

In summary, using **CP1** as a single-crystal platform, we have provided a new strategy for controlling the alignment of C $=$ C groups as a function of temperature, by fixing the regiospecific configurations of the intermediates and finally establishing the mechanism for the complex thermal/photoinduced reaction of **CP1** to **CP1-2α**. Going from **CP1** to **CP1-2α** requires multiple rotations of C $=$ C units and the cleavage and formation of C–C bonds. Furthermore, by setting different temperature

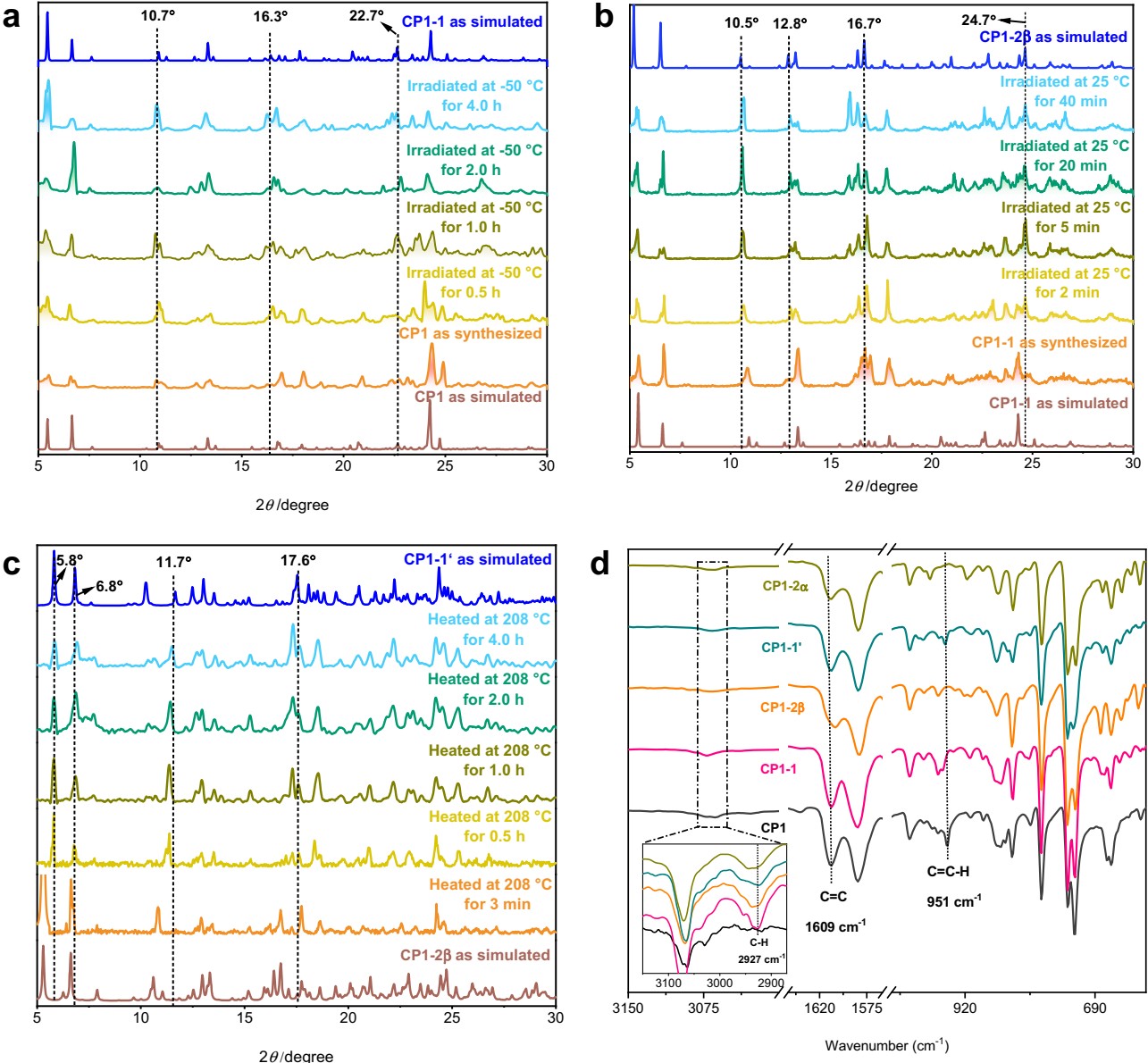

**Fig. 3 In-situ characterization. a** The in-situ PXRD patterns of the samples of **CP1** irradiated by UV light at −50 °C within different irradiation time intervals. **b** The in-situ PXRD patterns of the samples of **CP1-1** irradiated by UV light at 25 °C within different irradiation time intervals. **c** The in-situ PXRD patterns of the samples of **CP1-2β** heated at 208 °C within different time intervals. **d** FT-IR spectra of **CP1**, **CP1-1**, **CP1-2β**, **CP1-1′**, and **CP1-2α**.

regions for this thermal/photoinduced reaction, we could control the alignment of C=C groups of the F-1,3-bpeb ligands and trigger the formation of intermediates with special configurations. The whole process occurs in four steps with three isolated intermediates **CP1-1**, **CP1-2β**, and **CP1-1′**. In contrast to organic reactions occurring in solution via intermediates, which are often difficult to control, identify and capture, our strategy using **CP1** as a single-crystal platform clearly facilitates tracking, trapping and identification of reaction intermediates by in situ SCXRD, PXRD, IR, and other characterization techniques. DFT calculations provided a mechanistic picture of a series of transformations between **CP1** and **CP1-2α**. This approach is anticipated to open a door for the more predictable synthesis of cyclobutane compounds with specific configurations that cannot be isolated from photocycloaddition reactions in solution. Furthermore, it could be applied to disclose the possible configuration transformations of functional groups and intermediates and unravel the detailed mechanism for any inorganic, organic, and

macromolecular reactions, provided these reactions could be engineered into single-crystal coordination polymer platforms.

## Methods

**Synthesis of [Cd₂(F-1,3-bpeb)₂(3,5-DBB)₄] (CP1).** A thick Pyrex tube was loaded with 3CdSO₄·8H₂O (250.0 mg, 0.92 mmol), F-1,3-bpeb (6.0 mg, 0.021 mmol), 3,5-HDBB (11.7 mg, 0.042 mmol), 1.5 mL of DMF/H₂O (v/v = 1:4) and one drop of concentrated HNO₃. The tube was sealed and heated at 140 °C for 5 h. It was then cooled to room temperature to form colorless crystals of **CP1**, which were collected by filtration, washed with EtOH and H₂O, and dried in air. Yield: 90% (based on F-1,3-bpeb); ¹H NMR (400 MHz, DMSO-$d_6$): δ 8.60 (d, $J$ = 6.0 Hz, 4H), 8.04 (s, 4H), 7.95 (s, 2H), 7.79 (s, 2H), 7.60 (d, $J$ = 16.0 Hz, 2H), 7.59 (d, $J$ = 6.0 Hz, 4H), 7.52 (d, $J$ = 10.0 Hz, 2H), 7.44 (d, $J$ = 16.0 Hz, 2H); ¹⁹F NMR (377 MHz, DMSO-$d_6$): δ -113.13 ppm; IR (KBr): 3438, 1609, 1584, 1542, 1431, 1374, 1220, 1136, 1093, 1068, 1016, 951, 863, 848, 785, 738, 726, 661, 537, 512 cm⁻¹; analysis (calcd., found for C₆₈H₄₂Br₈CdF₂N₄O₈): C (44.56, 44.55), H (2.31, 2.30), N (3.06, 3.10).

**Synthesis of [Cd₂(1)(3,5-DBB)₄] (CP1-1).** The as-synthesized crystals of **CP1** were placed between glass slides and irradiated with an LED lamp (λ = 365 nm) for

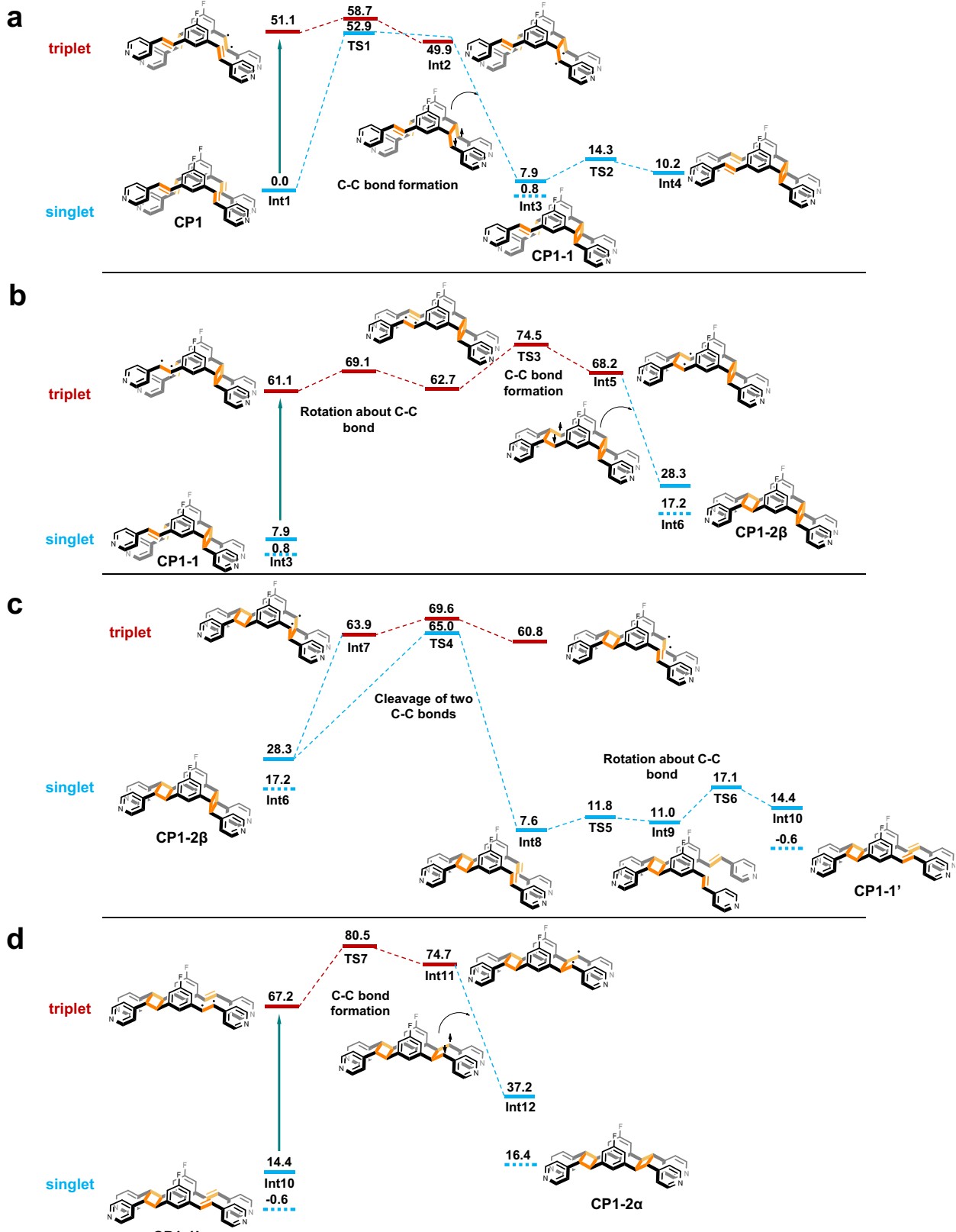

**Fig. 4 Calculated energy profiles. a–d** Reaction energy diagram (in kcal/mol) for the transformation processes from **CP1** to **CP1-1**, **CP1-1** to **CP1-2β**, **CP1-2β** to **CP1-1′**, and **CP1-1′** to **CP1-2α**. The thick bottle-green arrow signifies a step that requires photoexcitation. The thick blue dashed lines for **Int3**, **Int6**, **Int10**, and **Int12** are the relative energies obtained by using the respective X-ray structures when building the model in Supplementary Fig. 26.

approximately 4 h at −50 °C to form faint yellow crystals of **CP1-1** (100% yield based on **CP1**). $^1$H NMR (400 MHz, DMSO-$d_6$): δ 8.50 (d, $J$ = 6.0 Hz, 4H), 8.39 (d, $J$ = 6.0 Hz, 4H), 8.04 (s, 8H), 7.95 (s, 4H), 7.48 (m, 8H), 7.28 (d, $J$ = 6.0 Hz, 4H), 7.23 (d, $J$ = 16.0 Hz, 4H), 7.09 (d, $J$ = 9.6 Hz, 2H), 4.74 (d, $J$ = 6.8 Hz, 2H), 4.67 (d, $J$ = 6.8 Hz, 2H); $^{19}$F NMR (377 MHz, DMSO-$d_6$): δ -113.82 ppm; IR (KBr): 3438, 2927, 1608, 1590, 1541, 1432, 1376, 1221, 1138, 1102, 1068, 1017, 967, 951, 867, 836, 785, 739, 725, 662, 564, 511 cm$^{-1}$; analysis (calcd., found for C$_{68}$H$_{42}$Br$_8$CdF$_2$N$_4$O$_8$): C (44.56, 44.43), H (2.31, 2.25), N (3.06, 3.18).

**Synthesis of [Cd$_4$(2β)$_2$(3,5-DBB)$_8$] (CP1-2β).** The as-synthesized crystals of **CP1** or **CP1-1** were placed between glass slides and irradiated with an LED lamp (λ = 365 nm) for approximately 1 h at room temperature to form yellow crystals of **CP1-2β** (100% yield based on **CP1**). $^1$H NMR (400 MHz, DMSO-$d_6$): δ 8.43 (m, 8H), 8.04 (s, 8H), 7.98 (s, 4H), 7.32 (d, $J$ = 6.0 Hz, 4H), 7.27 (d, $J$ = 6.0 Hz, 4H), 7.17 (s, 2H), 6.71 (d, $J$ = 9.2 Hz, 2H), 6.46 (d, $J$ = 9.2 Hz, 2H), 4.84 (m, 8H); $^{19}$F NMR (377 MHz, DMSO-$d_6$) δ −114.88 ppm; IR (KBr): 3423, 2927, 1605, 1584, 1541, 1431, 1374, 1287, 1222, 1149, 1100, 1069, 1017, 968, 864, 836, 785, 739, 680, 661, 554, 491 cm$^{-1}$; analysis (calcd., found for C$_{136}$H$_{84}$Br$_{16}$Cd$_2$F$_4$N$_8$O$_{16}$): C (44.56, 44.33), H (2.31, 2.45), N (3.06, 3.13).

**Synthesis of [Cd$_2$(1′)(3,5-DBB)$_4$] (CP1-1′).** The crystals of **CP1-2β** were heated at 208 °C for approximately 4 h to form brown crystals of **CP1-1′** (100% yield based on **CP1-2β**). $^1$H NMR (400 MHz, DMSO-$d_6$): δ 8.50 (d, $J$ = 6.0 Hz, 4H), 8.39 (d, $J$ = 6.0 Hz, 4H), 8.04 (s, 8H), 7.95 (s, 4H), 7.48 (m, 8H), 7.28 (d, $J$ = 6.0 Hz, 4H), 7.23 (d, $J$ = 16.0 Hz, 4H), 7.09 (d, $J$ = 9.6 Hz, 2H), 4.74 (d, $J$ = 6.8 Hz, 2H), 4.67 (d, $J$ = 6.8 Hz, 2H); $^{19}$F NMR (377 MHz, DMSO-$d_6$): δ −113.82 ppm; IR (KBr): 3423, 2927, 1609, 1584, 1540, 1431, 1373, 1286, 1223, 1148, 1096, 1069, 1018, 967, 951, 859, 836, 785, 739, 661, 562, 509 cm$^{-1}$; analysis (calcd., found for C$_{68}$H$_{42}$Br$_8$CdF$_2$N$_4$O$_8$): C (44.56, 44.45), H (2.31, 2.35), N (3.06, 3.50).

**Synthesis of [Cd$_2$(2α)(3,5-DBB)$_4$] (CP1-2α).** The as-synthesized crystals of **CP1** were irradiated with an LED lamp (λ = 365 nm) for approximately 3 h at 208 °C to form brown crystals of **CP1-2α** (100% yield based on **CP1**). Alternatively, as-synthesized crystals of **CP1-1′** were irradiated with an LED lamp (λ = 365 nm) for approximately 1.5 h at room temperature to form brown crystals of **CP1-2α** (100% yield based on **CP1-1′**). $^1$H NMR (400 MHz, DMSO-$d_6$): δ 8.39 (d, $J$ = 6.0 Hz, 8H), 8.04 (s, 8H), 7.98 (s, 4H), 7.31 (d, $J$ = 6.0 Hz, 8H), 6.68 (d, $J$ = 10.0 Hz, 2H), 6.61 (s, 2H), 4.75 (d, $J$ = 6.0 Hz, 4H), 4.53 (d, $J$ = 6.0 Hz, 4H); $^{19}$F NMR (377 MHz, DMSO-$d_6$): δ −113.66 ppm; IR (KBr): 3431, 2927, 1608, 1585, 1543, 1431, 1374, 1281, 1223, 1148, 1099, 1067, 1018, 991, 980, 917, 864, 835, 784, 739, 677, 661, 554 cm$^{-1}$; analysis (calcd., found for C$_{68}$H$_{42}$Br$_8$CdF$_2$N$_4$O$_8$): C (44.56, 44.65), H (2.31, 2.45), N (3.06, 3.00).

**Photoirradiation experiments at different temperatures.** Single crystals of **CP1** were placed in a long glass tube or glass weighing bottle at different temperatures and irradiated with an LED lamp PLS-LED100B (λ = 365 nm, intensity ~500 mW cm$^{-2}$) for a period of time to form the photoproduct ca. 100% yield.

**Computational methods.** Density functional theory (DFT) calculations were employed to explore the reaction pathways of the isomerization reaction. Geometry optimization and frequency calculations were performed for the model illustrated below at the M06-2X/[Lanl2dz(Cd),6–31 G*(others)] level, and single-point energy calculations were carried out at the M06-2X/def2-TZVP level. Energy values obtained at M06-2X/[Lanl2dz(Cd),6–31 G*(others)] and the M06-2X/def2-TZVP levels are referred to as E1 and E2, respectively[49–55]. Zero-point energy (ZPE) values were also used to evaluate the relative energies of different species. Relative E2 + ZPE values were used in the energy diagrams. Gaussian 09 software[56] was used for DFT calculations.

## Data availability

Synthetic and experimental procedures, as well as crystallographic, NMR spectra, PXRD, FT-IR, and computational data are provided in the Supplementary Information. Crystallographic data for the structures reported in this article have been deposited at the Cambridge Crystallographic Data Centre, under deposition numbers CCDC 1889155 (**CP1**), 2036562 (**CP1-1**), 2036563 (**CP1-2β**), 2036564 (**CP1-1′**), 2036565 (**CP1-2α**), 1889156 (**2β**) and 1889157 (**2α**). Copies of the data can be obtained free of charge via https://www.ccdc.cam.ac.uk/structures/. All other data supporting the findings of this study are available within the article and its Supplementary Information. Data are also available from the corresponding author upon request. Source data are provided in this paper.

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

## Acknowledgements

We thank the National Natural Science Foundation of China (Grant nos. 21531006 and 21773163 to J.P.L.), the Collaborative Innovation Center of Suzhou Nano Science and Technology, the Priority Academic Program Development of Jiangsu Higher Education Institutions, the Project of Scientific and Technologic Infrastructure of Suzhou for financial support (Grant No. SZS201905 to J.P.L.) and a Presidential Fellowship and a University Development Fund (UDF01001996 to H.H.).

## Author contributions

M.F.W., F.L.H., Z.N. P.B. and J.P.L. conceived and designed the experiments. M.F.W. conceived and carried out experiments, determined structures, analyzed data, and corrected the draft of the paper. Y.M. assisted in the solvothermal synthesis and structural characterization of the title compounds and analyzed the data. H.H. performed DFT calculations. M.F.W., F.L.H., Z.N., P.B., and J.P.L. analyzed data and wrote the manuscript. All authors contributed to the discussion and revision of the paper.

## Competing interests

The authors declare no competing interests.
