## [Peer Review File · Nature Communications]

Controllable multiple-step configuration transformations in a thermal/photoinduced reactionReviewers' Comments:

Reviewer #1:

Remarks to the Author:

Although thermal control of photo-induced solid-state cycloaddition products has been demonstrated before (References 35 and 41), this work goes far beyond what has been achieved to date. As such, this manuscript describes an important step forward and it is suitable for publication in Nature Communications after minor revision.

The main findings of the work are nicely summarised in Figure 2, which provides a powerful demonstration of thermal control over a wide range of possible products for photoreaction between two diolefin compounds. The results described in the manuscript are fully supported by a comprehensive set of supplementary material, including all of the necessary crystal structures and details of other complementary analytical techniques. The main conclusions are also nicely supported by computational work. I fully appreciate the difficulties in obtaining suitable single crystals after significant 'abuse' in the form of irradiation and thermal treatment, so the crystallography is impressive despite some minor disorder issues.

Regarding any possible mechanisms, it is not easy to rule out that photodimerisation at one site may cause changes in the relative arrangement of the molecules, which may lead to favourable arrangement of the remaining site for photoreaction - a type of cooperative effect. Nevertheless, the authors have provided a convincing argument with regard to the intermediate stages of some of the multi-step reactions.

I have the following suggestions for minor revision of the text:

Page 5:

Replace "got changed" with "were altered" in the sentence "Interestingly, the original configuration of the monomer in the initial CP1 got changed at different temperature during the thermal/photoinduced reaction process."

Page 6:

Rewrite "According to Schmidt's rule, the distance between the two parallel C=C groups is 3.82 Å, which allows the photocycloaddition reaction." as "The distance between the two parallel C=C groups is 3.82 Å, which, according to Schmidt's rule, should allow the photocycloaddition reaction to occur."

Page 10:

Rewrite "Since a temperature increase can restore the C=C pairs by breaking cyclobutane rings, the transformation of CP1-2β to CP1-2α was investigated by in situ SCXRD at 208 °C, as determined by DSC measurements (Supplementary Fig. 24)." as "Since a temperature increase can restore the C=C pairs by breaking cyclobutane rings, as inferred from measurements (Supplementary Fig. 24), the transformation of CP1-2β to CP1-2α was investigated at 208 °C by in situ SCXRD."

Page 11:

Change "Upon heating the crystals of CP1-2β to 208 °C led to a decrease in the sharp pristine diffraction peaks of ..." to "Heating the crystals of CP1-2β to 208 °C led to a decrease in the sharp pristine diffraction peaks of ..."

Reviewer #2:

Remarks to the Author:

In this manuscript the authors describe the mechanism of the combined photochemical/thermal control of stereochemistry in a solid-state photochemical 2+2 reaction. The manuscript reads very

well, and the experimental part appears technically sound and solid. However, I find the topic too specialized and not of sufficiently broad appeal to be suitable for Nature.

In the computational part, which is a smaller part of the manuscript though, I also have a few critical remarks.

1) The authors assume the reaction to occur in the triplet state. Is this justified? 2+2 reactions are allowed photochemical reactions and should quickly proceed also in the singlet manifold.

2) Only ground state methods were employed to study excited states. For a comprehensive understanding TDDFT calculations are needed, singlet and triplet states calculated and the corresponding reaction pathways should be studied. For an efficient intersystem crossing, the spin-orbit couplings should be large and also calculated.

From the perspective of a computational chemist like me, the theory part is rather weak and premature. I can serve, if at all, as a weak support of the experimental findings, but not as a full fledged analysis of the reaction mechanism.

Point-by-point response to the reviewers' comments on MS ID NCOMMS-22-03066-T

To reviewer #1 (Remarks to the Author):

Comment 1: Although thermal control of photo-induced solid-state cycloaddition products has been demonstrated before (References 35 and 41), this work goes far beyond what has been achieved to date. As such, this manuscript describes an important step forward and it is suitable for publication in Nature Communications after minor revision.

The main findings of the work are nicely summarised in Figure 2, which provides a powerful demonstration of thermal control over a wide range of possible products for photoreaction between two diolefin compounds. The results described in the manuscript are fully supported by a comprehensive set of supplementary material, including all of the necessary crystal structures and details of other complementary analytical techniques. The main conclusions are also nicely supported by computational work. I fully appreciate the difficulties in obtaining suitable single crystals after significant 'abuse' in the form of irradiation and thermal treatment, so the crystallography is impressive despite some minor disorder issues.

Regarding any possible mechanisms, it is not easy to rule out that photodimerisation at one site may cause changes in the relative arrangement of the molecules, which may lead to favorable arrangement of the remaining site for photoreaction - a type of cooperative effect. Nevertheless, the authors have provided a convincing argument with regard to the intermediate stages of some of the multi-step reactions.

I have the following suggestions for minor revision of the text:

Response: We thank the reviewer for the very positive comments and support of our work.

Comment 2: Page 5: Replace "got changed" with "were altered" in the sentence "Interestingly, the original configuration of the monomer in the initial CP1 got changed at different temperature during the thermal/photoinduced reaction

process."

Response: We agree with the valuable suggestion of the reviewer. Per the reviewer's suggestion and formatting instructions, the sentence "Interestingly, ...in the initial CP1 got changed at different..." has been replaced by the sentence "Interestingly, ...in the initial **CP1** is altered at different ..." in the revised MS. (see Page 5, lines 4-6)

Comment 3: Page 6: Rewrite "According to Schmidt's rule, the distance between the two parallel C=C groups is 3.82 Å, which allows the photocycloaddition reaction." as "The distance between the two parallel C=C groups is 3.82 Å, which, according to Schmidt's rule, should allow the photocycloaddition reaction to occur."

Response: We agree with the valuable suggestion of the reviewer. Per the reviewer's suggestion, the sentence "According to Schmidt's rule, ...which allows the photocycloaddition reaction." has been replaced by the sentence "The distance between the two parallel C=C groups is 3.82 Å, which, according to Schmidt's rule, should allow the photocycloaddition reaction to occur." in the revised MS. (see Page 6, bottom to lines 1-3)

Comment 4: Page 10: Rewrite "Since a temperature increase can restore the C=C pairs by breaking cyclobutane rings, the transformation of CP1-2 β to CP1-2 α was investigated by in situ SCXRD at 208 °C, as determined by DSC measurements (Supplementary Fig. 24)." as "Since a temperature increase can restore the C=C pairs by breaking cyclobutane rings, as inferred from measurements (Supplementary Fig. 24), the transformation of CP1-2 β to CP1-2 α was investigated at 208 °C by in situ SCXRD."

Response: We agree with the valuable suggestion of the reviewer. As suggested, the sentence "Since a temperature increase can restore the C=C pairs by breaking cyclobutane rings, the transformation of CP1-2 β to CP1-2 α was investigated by in situ SCXRD at 208 °C, as determined by DSC measurements (Supplementary Fig. 24)." has been replaced by the sentence "Since a temperature increase can restore the C=C pairs by breaking cyclobutane rings, as inferred from measurements (Supplementary Fig. 24), the transformation of **CP1-2 β** to **CP1-2 α** was investigated at 208 °C by in situ SCXRD." (see Page 10, lines 9-12 in the revised MS)

Comment 5: Page 11: Change "Upon heating the crystals of CP1-2 β to 208 °C led to a decrease in the sharp pristine diffraction peaks of ..." to "Heating the crystals of CP1-2 β to 208 °C led to a decrease in the sharp pristine diffraction peaks of ..."

Response: We agree with the valuable suggestion of the reviewer. As suggested, the sentence "Upon heating the crystals of CP1-2 β to 208 °C led to a decrease in the sharp pristine diffraction peaks of ..." has been replaced by the sentence "Heating the crystals of CP1-2 β to 208 °C led to a decrease in the sharp pristine diffraction peaks of ..." (see Page 11, bottom to lines 7-8 in the revised MS)

To reviewer #2 (Remarks to the Author):

Comment: In this manuscript the authors describe the mechanism of the combined photochemical/thermal control of stereochemistry in a solid-state photochemical 2+2 reaction. The manuscript reads very well, and the experimental part appears technically sound and solid. However, I find the topic too specialized and not of sufficiently broad appeal to be suitable for Nature.

Response: We thank the reviewer for the comments. In this work, we focused on the synergistic control of configuration transformations in a single-crystal coordination polymer platform, which represents a significant work in synthetic chemistry.

Photochemical [2+2] cycloaddition reaction is a green and powerful strategy that has been widely applied in diverse fields, ranging from drug discovery to materials engineering, because it generates specific molecules that would otherwise be difficult to access. As mentioned in this MS, various supramolecular templates, host-guest assemblies, quantum dots, chiral molecular catalysts, and Lewis acid cocatalysts have been developed to control the enantioselectivity and diastereoselectivity of [2+2] photocycloaddition transformations. However, the reported synthetic methods mainly focused on the stereochemical control of monocyclic products in the [2+2] photocycloaddition reaction, which seems less effective to synthesize a set of multiple-cyclic isomers with a controlled selectivity and specificity through one [2+2] photocycloaddition reaction. In addition, in contrast to thermal cycloaddition reactions, the intermediate species of a [2+2] photocycloaddition is short-lived and requires efficient trapping techniques. Although

it is possible to capture cyclic intermediates at low temperature, the control of their stereochemistry during the reaction has not yet been documented.

Single-crystal CPs appear to be potentially powerful platforms for tracking the stereo-chemical structure of reaction intermediates. There are some papers about the “Single-crystal platforms”. For example, Fujita and coworkers have experimentally shown that some porous MOFs crystals can be used as a crystal sponge for trapping molecules that are difficult to be isolated or as a crystal container/capsule for running special reactions (see *Nature*, **495**, 461-466 (2013); *Angew. Chem. Int. Ed.* **58**, 9171-9173 (2019); **60**, 11809-11813 (2021)). Different from these papers, we are interested in synergistic control of configuration transformations in a single-crystal coordination polymer platform, and tracing the changes in the configuration of the isomers in the synergistic thermal/photoinduced reactions using SCXRD and other techniques. Therefore, we report a unique approach to control the regiospecific configurations of C=C groups and the intermediates by varying temperatures in multiple-step thermal/photoinduced reactions using a single-crystal coordination polymer platform. This approach could be generalized to the analysis of the possible configuration transformations of functional groups and intermediates and unravel the detailed mechanism for any inorganic, organic and macromolecular reactions susceptible to incorporation into single-crystal coordination polymer platforms.

Comment: In the computational part, which is a smaller part of the manuscript though, I also have a few critical remarks.

The authors assume the reaction to occur in the triplet state. Is this justified? 2+2 reactions are allowed photochemical reactions and should quickly proceed also in the singlet manifold.

Response: First of all, we sincerely thank the reviewer for the valuable comments. The S_1 singlet state is usually short-lived compared with the T_1 state, and many [2+2] photocycloaddition reactions have been explained in terms of triplet chemistry (e.g., Lautfy, R. O. & de Mayo, P. *Can. J. Chem.* **50**, 3465-3471 (1972) & many papers including the recent papers cited in the main text). In addition, our coordination polymer contains heavy cadmium ions. For these reasons, we assumed that the triplet state is a more likely spin state for the photocycloaddition. However, as pointed out by the reviewer, we cannot completely rule out the possibility that the

singlet manifold is used, and which spin state is chosen is sometimes of critical importance. For example, Bach and coworkers (Brimioulle, R., Guo, H. & Bach, T. *Chem. Eur. J.* **18**, 7552-7560 (2012); Brimioulle, R., Bauer, A. & Bach, T. *J. Am. Chem. Soc.* **137**, 5170-5176 (2015)) showed that the enantioselectivity of [2+2] reactions of coumarin substrates is altered depending on which spin state is used. Chen, Dolg, and coworkers (Wang, H. J., Cao, X. Y., Chen, X. B., Fang, W. H & Dolg, M. *Angew. Chem., Int. Ed.* **54**, 14295-14298 (2015); Wang, H. J., Fang, F. H. & Chen, X. B. *J. Org. Chem.* **81**, 7093-7101 (2016)) performed extensive *ab initio* calculations to rationalize the observed enantioselectivity. However, the geometric conditions of the linkers in our system do not cause this sort of issue, irrespective of the spin state used.

Therefore, we have modified the original descriptions as follows:

"As shown in Fig. 4a, the barrier for the thermal C–C bond formation of **Int1(S0)** via **TS1(S0)** having an open-shell character is 52.9 kcal/mol, which is too high. Therefore, **Int1(S0)** will first undergo photoexcitation, and the C–C bond formation may occur on the T_1 energy surface, to form **Int3 (CP1-1)**. Given that $^1\text{Int3}$ is thermodynamically favoured over $^1\text{Int4}$, a likely pathway involves photoexcitation of $^1\text{Int3}$, and the rotation and C–C bond formation events then occur in the triplet state, before forming $^1\text{Int6 (CP1-2}\beta\text{)}$ (Fig. 4b)." (see Page 13, bottom to lines 1-7 in the revised MS)

"If this C–C bond cleavage event occurs in the triplet state, the process would involve spin inversions. Furthermore, $^3\text{TS4}$ is less stable than $^1\text{TS4}$. These factors make it unlikely that the triplet pathway is used under thermal conditions." (see Page 15, lines 7-10 in the revised MS)

"DFT calculations provided a mechanistic picture of a series of transformations between **CP1** and **CP1-2 α** ." (see Page 16, lines 5-6 in the revised MS)

Also, to make the issue of the spin state clearer, we have amended the following sentences in the newly added Section 11.3 in the Supplementary Information.

"Singlet excited states are typically short-lived, and our coordination polymer contains heavy cadmium ions. As such, we assumed that [2+2] photocycloaddition reactions occur in the T_1 spin state, which may be generated from a photoexcitation to a singlet excited state and a subsequent intersystem crossing event. Indeed, the

triplet spin state is invoked in the majority of known [2+2] photocycloaddition reactions¹¹. It should be noted that there are also cases where the singlet manifold is preferentially used for [2+2] photocycloaddition, especially when heavy elements are not involved. Which excited-state pathway is chosen during cycloaddition sometimes has a critical impact on the product selectivity. For example, Bach and coworkers have experimentally shown that [2+2] photocycloaddition reactions use different spin states and exhibit different enantioselectivity depending on the reaction condition^{12,13}, and Chen, Dolg, and coworkers have performed ab initio theoretical studies to rationalize the experimental results^{14,15}. However, the geometric features and constraints in our coordination polymer preclude the spin-dependent induction of enantioselectivity.” (see Pages S46-S47)

We have added the following references to the Supplementary References:

11. Lautfy, R. O. & de Mayo, P. Primary bond formation in the addition of cyclopentenone to chloroethylenes. *Can. J. Chem.* **50**, 3465-3471 (1972). Also see the papers cited in the main text.
12. Brimioulle, R., Guo, H. & Bach, T. Enantioselective intramolecular [2+2] photocycloaddition reactions of 4-substituted coumarins catalyzed by a chiral Lewis acid. *Chem. Eur. J.* **18**, 7552-7560 (2012).
13. Brimioulle, R., Bauer, A. & Bach, T. Enantioselective Lewis acid catalysis in intramolecular [2+2] photocycloaddition reactions: a mechanistic comparison between representative coumarin and enone substrates. *J. Am. Chem. Soc.* **137**, 5170-5176 (2015).
14. Wang, H. J., Cao, X. Y., Chen, X. B., Fang, W. H & Dolg, M. Regulatory mechanism of the enantioselective intramolecular enone [2+2] photocycloaddition reaction mediated by a chiral Lewis acid catalyst containing heavy atoms. *Angew. Chem., Int. Ed.* **54**, 14295-14298 (2015).
15. Wang, H. J., Fang, F. H. & Chen, X. B. Mechanism of the enantioselective intramolecular [2+2] photocycloaddition reaction of coumarin catalyzed by a chiral Lewis acid: comparison with enone substrates. *J. Org. Chem.* **81**, 7093-7101 (2016).

Comment: Only ground state methods were employed to study excited states. For a

comprehensive understanding TDDFT calculations are needed, singlet and triplet states calculated and the corresponding reaction pathways should be studied. For an efficient intersystem crossing, the spin-orbit couplings should be large and also calculated.

Response: We thank the reviewer for the comment and valuable suggestion. If our main conclusion is that the reaction proceeds exclusively in the triplet state but not in the singlet state to yield a specific product, careful excited-state calculations are certainly needed. This is because the reaction is impossible in that case without sufficiently large spin-orbit coupling. However, we do not intend to argue it. We just assume that the triplet state is used for the above-mentioned reasons, and even if the singlet state is used, the reaction product is not altered.

We would also like to point out that there are several technical concerns in TDDFT calculations, especially for the application to our coordination polymer system.

1. First of all, the computational demands of the TDDFT reaction-pathway calculations with geometry optimization calculations are prohibitively high. As we have >100 atoms, geometry optimization of intermediates in excited states is already too demanding, and Gaussian 09 cannot calculate analytic 2nd-order derivatives for transition-state geometry optimization.
2. An alternative strategy may be employed to calculate excited-state energies by single-point calculations using the ground-state geometries. However, the singlet ground state has a significant open-shell character around **TS1** (see new Fig. 4a). Therefore, TDDFT-calculated triplet excited states suffer from severe spin contamination compared to triplet **TS1** in the ground-state calculation. Thus, we cannot expect significant improvement in the description of the T₁ excited state by performing demanding TDDFT calculations for our system.

Comment: From the perspective of a computational chemist like me, the theory part is rather weak and premature. I can serve, if at all, as a weak support of the experimental findings, but not as a full fledged analysis of the reaction mechanism.

Response: We thank the reviewer for the comment. We would like to humbly make the following comments. After doing computational studies, we realized that the hypothetical mechanism that we had long ago was completely different from the mechanism presented in the manuscript (and was not very reasonable in hindsight).

The computational study indeed has played a crucial role in this work, especially in delineating the mechanistic picture. We have also amended detailed descriptions of our computational study to the Supplementary Information, which we hope will minimize potential misunderstandings.

Reviewers' Comments:

Reviewer #2:

Remarks to the Author:

The authors have addressed my scientific comments sufficiently and comprehensively, however, I still find the paper too specialized for Nature. The latter decision however lies with the editors.

Response to referees and editors

To reviewer #2 (Remarks to the Author)

Comment 1: The authors have addressed my scientific comments sufficiently and comprehensively, however, I still find the paper too specialized for Nature. The latter decision however lies with the editors.

Response: We thank the reviewer for the comments. We highly appreciate that the editor made the very positive decision to accept this article. Our paper highlights the successful configuration control of C=C pairs and of the intermediates by varying temperatures, thus allowing to understand the detailed mechanism of a multi-step thermal/photo-induced reaction by using a single-crystal coordination polymer platform. Such a platform facilitates tracking, trapping and identification of reaction intermediates by e.g. *in-situ* SCXRD, PXRD, and IR techniques accompanied by DFT calculations deciphering the reaction mechanism for the entire process. This protocol opens a door for the more predictable photosynthesis of cyclobutane compounds with specific configurations that cannot be isolated in solution and unravels the detailed reaction mechanisms for any inorganic, organic and macromolecular reactions incorporated into single-crystal coordination polymer platforms. It will particularly attract a broad readership for the researchers focusing their research on the mechanisms of inorganic, organic, macromolecular reactions, supramolecular assembly, etc.

To the editor:

Your manuscript entitled "Controllable multiple-step configuration transformations in a thermal/photoinduced reaction" has now been seen again by our referees, whose comments appear below. In light of their advice I am delighted to say that we are happy, in principle, to publish a suitably revised version in Nature Communications under the open access CC BY license (Creative Commons Attribution 4.0 International License).

Comment 1: We therefore invite you to revise your paper one last time to address the remaining concerns of our reviewers and our editorial requests in the attached document(s). At the same time we ask that you edit your manuscript to comply with our policies and formatting requirements and to maximise the accessibility and therefore the impact of your work.

Please see the attached document(s), listing a number of points that must be addressed. Failure to comply with our editorial requests will cause delays in accepting your manuscript. Please also see the Nature Communications formatting instructions for further information.

Response: We highly appreciate your very positive comment on our article. We have made a point-by-point response to your comments (see below) and carefully revised the manuscript accordingly. The corresponding changes have been highlighted in yellow in the main text.

SUBMISSION INFORMATION

In order to accept your paper, we require the following:

Comment 2: - A revised author checklist describing your response to our editorial requests (attached).

Response: We have made a point-by-point response to editorial requests in the revised author checklist and carefully revised the manuscript accordingly.

Comment 3: - A separate point-by-point response to the reviewers' comments, reproduced verbatim.

Response: We have made a response to the reviewer's comment (see below).

Comment 4: - The final version of your manuscript as a Word or LaTeX file, with all changes highlighted in the text and any tables prepared using the table menu in Word or the table environment in LaTeX.

Response: The corresponding changes have been highlighted in yellow in the manuscript.

Comment 5: - If using LaTeX, please use numerical references only for citations, and include the references within the manuscript file itself. If you wish to use BibTeX, please copy the reference list from the .bbl file, paste it into the main manuscript .tex file, and delete the associated `\bibliography` and `\bibliographystyle` commands.

Response: The final version of our manuscript was a Word file, not a LaTeX or BibTeX file.

Comment 6: - The complete author list provided in the manuscript file, which must match that given on our manuscript tracking system. The author list in the main manuscript file will be used during typesetting of your article.

Response: We have checked it, and the author list in the manuscript file has been consistent with that provided in the manuscript tracking system.

Comment 7: - Production-quality versions of each figure as a separate file containing all panels. To ensure the swift processing of your paper, please provide the highest quality versions of your images and when combining different figure parts into one file for layout, use a vector-

based application such as Adobe Illustrator or Microsoft Powerpoint. We recommend .ai, .eps, .pdf, .ppt. Figures divided into panels should be labelled with a lower-case, boldface 'a', 'b', etc. in the top left-hand corner. If resolution is not of sufficient quality, production of your paper will be held whilst replacement files are obtained. For detailed guidance on figure preparation, see <https://www.nature.com/documents/aj-artworkguidelines.pdf>

Response: According to the guidelines, production-quality versions of four figures in MS have been provided in the separate PPT files.

Comment 8: - Please note that we do not modify the text in figures to conform to style during the production process. Please ensure that your figures are presented accurately and adhere to the guidance provided.

Response: We ensure that our figures are presented accurately and adhere to the guidance provided.

Comment 9: - Any updated checklists that verify compliance with our research ethics and data reporting standards in PDF format.

Response: The format of updated checklists has already followed this guidance.

Comment 10: - The final version of the Supplementary Information in one PDF file.

Response: We have re-converted the Supplementary Information into one PDF file.

Comment 11: - Any Supplementary Movie, Audio, Data and Software submitted as separate files. Supplementary Data and Source Data must be provided as .xls, .xlsx or .zip files, while Supplementary Software must be supplied as .zip files.

Response: There is no Supplementary Movie/Audio/Data file in our article. The source data has been provided in a single Excel file for each figure in a separate sheet within a zipped folder. Name this folder 'Source Data', and include the statement "Source data are provided with this paper." in "Data Availability" section.

Comment 12: ** Please note that we do not edit Supplementary Information files; they must be finalised prior to acceptance of the paper. **

Response: We have finished the modification of the Supplementary Information.

Comment 13: - If you wish, an interesting image (but not an illustration or schematic) for consideration as a Featured Image on the Nature Communications homepage. The file should be 1200x675 pixels in RGB format and should be uploaded as a Related Manuscript File. In

addition to our home page, we may also use this image (with credit) in other journal-specific promotional material.

- Completed and signed copies of our Multimedia License to Publish (LTP) for any Featured Image suggestions (please use one form for each image and give a scientific description of the image in the 'title' field; do not use "Featured Image" as a title): <http://www.nature.com/documents/snl-multimedia-ltp.docx>

Response: We will not provide Featured Image of this article on the Nature Communications homepage.

OPEN ACCESS

Comment 14: Nature Communications is a fully open access journal. Articles are made freely accessible on publication under a CC BY license (Creative Commons Attribution 4.0 International License). This license allows maximum dissemination and re-use of open access materials and is preferred by many research funding bodies.

For further information about article processing charges, open access funding, and advice and support from Nature Portfolio, please visit <http://www.nature.com/ncomms/about/open-access>

At acceptance, you will be provided with instructions for completing this CC BY license on behalf of all authors. This grants us the necessary permissions to publish your paper.

Response: We understand it and will provide this CCBY license according to the instructions at acceptance.

ORCID

Comment 15: Nature Communications is committed to improving transparency in authorship. As part of our efforts in this direction, we are now requesting that all authors identified as 'corresponding author' create and link their Open Researcher and Contributor Identifier (ORCID) with their account on the Manuscript Tracking System (MTS) prior to acceptance. ORCID helps the scientific community achieve unambiguous attribution of all scholarly contributions. For more information please visit <http://www.springernature.com/orcid>

For all corresponding authors listed on the manuscript, please follow the instructions in the link below to link your ORCID to your account on our MTS before submitting the final version of the manuscript. If you do not yet have an ORCID you will be able to create one in minutes.

IMPORTANT: All authors identified as 'corresponding author' on the manuscript must follow these instructions. Non-corresponding authors do not have to link their ORCIDs but are encouraged to do so. Please note that it will not be possible to add/modify ORCIDs at proof. Thus, if they wish to have their ORCID added to the paper they must also follow the above procedure prior to acceptance.

To support ORCID's aims, we only allow a single ORCID identifier to be attached to one account. If you have any issues attaching an ORCID identifier to your MTS account, please contact the Platform Support Helpdesk.

Response: Following these instructions, all corresponding authors of this article have created and linked their ORCID with their account on the MTS prior to acceptance.